# Frame Interpolation with Consecutive Brownian Bridge Diffusion

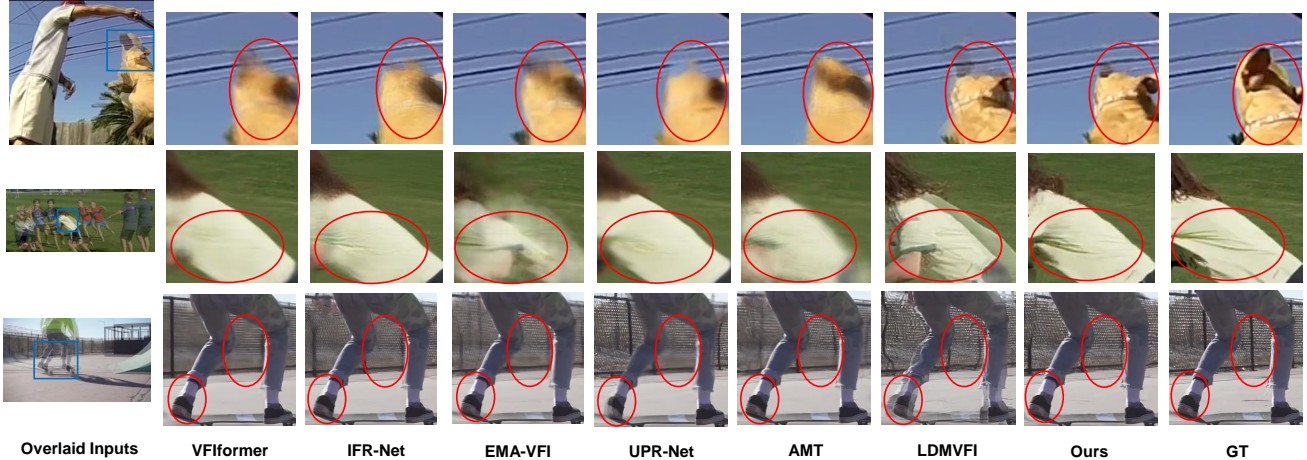

**Figure 1: Qualitative Comparison of our proposed method and recent state-of-the-art methods. Overlaid Inputs are the average of two neighboring frames, and the results predict their intermediate frame. Our method generates better and clearer interpolation results than recent SOTAs, such as clearer dog skins (first row), clearer cloth with folds (second row), and clearer fences with nets and high-quality shoes (third row). Images within blue boxes are displayed to better compare detailed qualities, and red circles highlight our better performance. Examples are chosen from SNU-FILM [8] extreme subset which is the hardest one with large motion changes. More visual results are provided in the supplementary materials.**

## ABSTRACT

Recent work in Video Frame Interpolation (VFI) tries to formulate VFI as a diffusion-based conditional image generation problem, synthesizing the intermediate frame given a random noise and neighboring frames. Due to the relatively high resolution of videos, Latent Diffusion Models (LDMs) are employed as the conditional generation model, where the autoencoder compresses images into latent representations for diffusion and then reconstructs images from these latent representations. Such a formulation poses a crucial challenge: VFI expects that the output is *deterministically* equal to the ground truth intermediate frame, but LDMs *randomly* generate a diverse set of different images when the model runs multiple times. The reason for the diverse generation is that the cumulative variance (variance accumulated at each step of generation) of generated latent representations in LDMs is large. This makes the sampling trajectory random, resulting in diverse rather than deterministic generations. To address this problem, we propose our unique solution: Frame Interpolation with Consecutive Brownian Bridge Diffusion. Specifically, we propose consecutive Brownian Bridge diffusion that takes a deterministic initial value as input, resulting in a much smaller cumulative variance of generated latent representations. Our experiments suggest that our method can improve together with the improvement of the autoencoder and achieve state-of-the-art performance in VFI, leaving strong potential for further enhancement.

## CCS CONCEPTS

• **Computing methodologies → Computer vision**.

## KEYWORDS

Video Frame Interpolation, Diffusion Models, Brownian Bridge

## 1 INTRODUCTION

Video Frame Interpolation (VFI) aims to generate high frame-per-second (fps) videos from low fps videos by estimating the intermediate frame given its neighboring frames. High-quality frame interpolation contributes to other practical applications such as novel view synthesis [14], video compression [58], and high-fps cartoon synthesis [47].

Current works in VFI can be divided into two folds in terms of methodologies: flow-based methods [1, 7, 12, 18, 20, 24, 29, 32, 34, 39, 42, 47, 60] and kernel-based methods [4, 5, 9, 27, 36, 37, 46]. Flow-based methods compute flows in the neighboring frames and forward warp neighboring images and features [18, 24, 34, 35, 47] or estimate flows from the intermediate frame to neighboring frames and backward warp neighboring frames and features [1, 7, 12, 20, 29, 32, 39, 42, 60]. Instead of relying on optical flows, kernel-based

methods predict convolution kernels for pixels in the neighboring frames. Recent advances in flow estimation [19, 21–23, 51, 52, 57] make it more popular to adopt flow-based methods in VFI.

Other than these two folds of methods, MCVD [55] and LD-MVFI [11] start formulating VFI as a diffusion-based image generation problem. LDMVFI considers VFI as a conditional generation task with Latent Diffusion Models (LDMs) [43], where LDMs contain an autoencoder that compresses images into latent representations and reconstructs images from latent representations. Diffusion models [17] run in the latent space of the autoencoder. Though diffusion models achieve excellent performance in image generation, there remain challenges in applying them to VFI.

(1) The formulation of diffusion models results in a large cumulative variance (the variance accumulated during sampling) of generated latent representations. The sampling process starts with standard Gaussian noise and adds small Gaussian noise to the denoised output at each step based on a pre-defined distribution. After the sampling process, images are generated, but these noises also add up to a large cumulative variance. Though such a variance is beneficial to diversity (i.e. repeated sampling results in *different* outputs), VFI requires that repeated sampling returns *identical* results, which is the ground truth intermediate frame. Therefore, a small cumulative variance is preferred in VFI. The relation of the cumulative variance and diversity is supported by the fact that DDIM [48] tends to generate relatively deterministic images than DDPM [17]. DDIM removes small noises at each sampling step, so the cumulative variance in DDIM is lower. LDMVFI [11] uses conditional generation as guidance, but this does not change the nature of large cumulative variance. In Section 3.4, we show that our method has a much lower cumulative variance than conditional generation.

(2) Videos usually have high resolution, which can be up to 4K [41], resulting in practical constraints to apply diffusion models [17] in pixel spaces. It is natural to apply Latent Diffusion Models (LDMs) [43] to sample latent representations and reconstruct them back to images. LDMs apply VQModels in VQGAN [13] to compress images into latent representations and reconstruct images from latent representations. However, it does not take advantage of neighboring frames, which can be a good guide to reconstruction. LDMVFI designs reconstruction models that leverage neighboring frames, but it tends to reconstruct overlaid images when there is a relatively large motion between neighboring frames, possibly due to the cross-attention with features of neighboring frames, which is shown in Figure 1.

To tackle these challenges, we propose a consecutive Brownian Bridge diffusion model (in latent space) that transits among three deterministic endpoints for VFI. This method results in a much smaller cumulative variance, achieving a better estimation of the ground truth inputs. We can separate LDM-based VFI methods into two parts: autoencoder and ground truth estimation (with diffusion). It is different from the original LDMs [43] because the latent representation generated by diffusion does not aim to estimate some ground truth. It is also different from LDMVFI [11] because LD-MVFI does not consider the performance of autoencoder separately from the interpolation method. With such a two-stage separation, we evaluate them separately for specific directions of improvement.

Moreover, we take advantage of flow estimation and refinement methods in recent literature [32] to improve the autoencoder. The feature pyramids from neighboring frames are warped based on estimated optical flows, aiming to alleviate the issues of reconstructing overlaid images. In experiments, our method improves by a large margin when the autoencoder is improved and achieves state-of-the-art performance. Our contribution can be summarized in three parts:

- We propose a new consecutive Brownian Bridge diffusion model for VFI and justify its advantages over traditional diffusion models: lower cumulative variance and better ground truth estimation capability. Additionally, we provide a cleaner formulation of Brownian Bridges and also propose the loss weights among different times in Brownian Bridges.
- We formulate the diffusion-based VFI as two stages: autoencoder and ground truth estimation. This is a novel interpretation of LDM-based VFI, which can provide specific directions for improvements.
- Through extensive experiments, we validate the effectiveness of our method. Our method estimates the ground truth better than traditional diffusion with conditional generation. Moreover, the performance of our method improves when the autoencoder improves and achieves state-of-the-art performance with a simple yet effective autoencoder, indicating its strong potential in VFI.

## 2 RELATED WORKS

### 2.1 Video Frame Interpolation

Video Frame Interpolation can be roughly divided into two categories in terms of methodologies: flow-based methods [1, 7, 12, 18, 20, 24, 29, 32, 34, 39, 42, 47, 60] and kernel-based methods [4, 5, 9, 27, 36, 37, 46]. Flow-based methods assume certain motion types, where a few works assume non-linear types [7, 12] while others assume linear. Via such assumptions, flow-based methods estimate flows in two ways. Some estimate flows from the intermediate frame to neighboring frames (or the reverse way) and apply backward warping to neighboring frames and their features [1, 7, 12, 20, 29, 32, 39, 42, 60]. Others compute flows among the neighboring frames and apply forward splatting [18, 24, 34, 35, 47]. In addition to the basic framework, advanced details such as recurrence of inputs with different resolution level [24], cross-frame attention [60], and 4D-correlations [29] are proposed to improve performance. Kernel-based methods, introduced by [36], aim to predict the convolution kernel applied to neighboring frames to generate the intermediate frame, but it has difficulty in dealing with large displacement. Following works [5, 9, 27] alleviate such issues by introducing deformable convolution. LDMVFI [11] recently introduced a method based on Latent Diffusion Models (LDMs) [43], formulating VFI as a conditional generation task. LDMVFI uses an autoencoder introduced by LDMs to compress images into latent representations, efficiently run the diffusion process, and then reconstruct images from latent space. Instead of directly predicting image pixels during reconstruction, it takes upsampled latent representations in the autoencoder as inputs to predict convolution kernels in kernel-based methods to complete the VFI task.

## 2.2 Diffusion Models

The diffusion model is introduced by DDPM [17] to image generation task and achieves excellent performance in high-fidelity and high-diversity image generation. The whole diffusion model can be split into a forward diffusion process and a backward sampling process. The forward diffusion process is defined as a Markov Chain with steps $t = 1, ..., T$, and the backward sampling process aims to estimate the distribution of the reversed Markov chain. The variance of the reversed Markov chain has a closed-form solution, and and expected value of the reversed Markov chain is estimated with a deep neural network. Though achieving strong performance in image generation tasks, DDPM [17] requires $T = 1000$ iterative steps to generate images, resulting in inefficient generation. Sampling steps cannot be skipped without largely degrading performance because the conditional distribution at step $t - 2$ needs to be computed with the conditional distribution at time $t - 1$ and $t$ due to its Markov property. To enable efficient and high-quality generation, DDIM [48] proposes a non-Markov formulation of diffusion models, where the conditional distribution at time $t - k$ ($k > 0$) can be directly computed with the conditional distribution at time $t$. Therefore, skipping steps does not largely degrade performance. Score-based SDEs [3, 49, 63] are also proposed as an alternative formulation of diffusion models by writing the diffusion process in terms of Stochastic Differential Equations [38], where the reversed process has a closed-form continuous time formulation and can be solved with Eluer's method with a few steps [49]. In addition, Probability Flow ODE is proposed as the deterministic process that shares the same marginal distribution with the reversed SDE [49]. Following score-based SDEs, some works propose efficient methods to estimate the solution Probability Flow ODE [30, 31]. Instead of focusing on the nature of the diffusion process, DeepCache [33] proposes a feature caching and sharing mechanism in the denoising UNet, enabling parallel and skipping computation and further improving efficiency. To deal with high-resolution images, the Latent Diffusion Model [43] proposes an autoencoder with a Vector Quantization Layer (VQ Layer) that compresses and reconstructs images, and diffusion models run with compressed images. With such an autoencoder, high-resolution images can be generated efficiently. Other than accelerating generation, diffusion models are applied to conditional generation tasks [3, 6, 28, 43, 45, 61, 63] such as generation based on poses or skeletons, image inpainting, etc.

## 3 METHODOLOGY

In this section, we will first go through preliminaries on the Diffusion Model (DDPM) [17] and Brownian Bridge Diffusion Model (BBDM) [28] and introduce the overview of our two-stage formulation: autoencoder and ground truth estimation (with consecutive Brownian Bridge diffusion). Then, we will discuss the details of our autoencoder method. Finally, we propose our solution to the frame interpolation task: consecutive Brownian Bridge diffusion.

## 3.1 Preliminaries

**Diffusion Model.** The forward diffusion process of Diffsuion Model [17] is defined as:

$$q(\mathbf{x}_t|\mathbf{x}_{t-1}) = \mathcal{N}(\mathbf{x}_t; \sqrt{1 - \beta_t}\mathbf{x}_{t-1}, \beta_t \mathbf{I}). \tag{1}$$

When $t = 1$, $\mathbf{x}_{t-1} = \mathbf{x}_0$ is a sampled from the data (images). By iterating Eq. (1), we get the conditional marginal distribution of $\mathbf{x}_t$ [17]:

$$q(\mathbf{x}_t|x_0) = \mathcal{N}(x_t; \sqrt{\alpha_t}\mathbf{x}_0, (1 - \alpha_t)\mathbf{I}), \tag{2}$$

$$\text{where } \alpha_t = \prod_{s=1}^{t}(1 - \beta_s).$$

The sampling process can be derived with the Bayes' theorem [17]:

$$p_\theta(\mathbf{x}_{t-1}|\mathbf{x}_t) = q(\mathbf{x}_{t-1}|\mathbf{x}_0, \mathbf{x}_t) = \mathcal{N}(x_{t-1}; \tilde{\mu}_t, \tilde{\beta}_t), \tag{3}$$

$$\text{where } \tilde{\mu}_t = \frac{\sqrt{\alpha_{t-1}}\beta_t}{1 - \alpha_t}\mathbf{x}_0 + \frac{\sqrt{1 - \beta_t}(1 - \alpha_{t-1})}{1 - \alpha_t}\mathbf{x}_t, \tag{4}$$

$$\text{and } \tilde{\beta}_t = \frac{1 - \alpha_{t-1}}{1 - \alpha_t}\beta_t. \tag{5}$$

Eq. (4) can be rewritten with Eq. (2) via reparameterization:

$$\tilde{\mu}_t = \frac{1}{1 - \beta_t}\left(\mathbf{x}_t - \frac{\beta_t}{\sqrt{1 - \alpha_t}}\epsilon\right), \text{ where } \epsilon \sim \mathcal{N}(0, \mathbf{I}). \tag{6}$$

By Eq. (4) and (6), we only need to estimate $\epsilon$ to estimate $p_\theta(\mathbf{x}_{t-1}|\mathbf{x}_t)$. Therefore, the training objective is:

$$\mathbb{E}_{\mathbf{x}_0, \epsilon}\left[||\epsilon_\theta(\mathbf{x}_t, t) - \epsilon||_2^2\right]. \tag{7}$$

It suffices to train a neural network $\epsilon_\theta(\mathbf{x}_t, t)$ predicting $\epsilon$.

**Brownian Bridge Diffusion Model.** Brownian Bridge [44] is a stochastic process that transits between two fixed endpoints, which is formulated as $X_t = W_t|(W_{t_1}, W_{t_2})$, where $W_t$ is a standard Wiener process with distribution $\mathcal{N}(0, t)$. We can write a Brownian Bridge as $X_t = W_t|(W_0, W_T)$ to define a diffusion process. When $W_0 = a, W_T = b$, it follows a normal distribution:

$$X_t \sim \mathcal{N}\left(\left(1 - \frac{t}{T}\right)a + \frac{t}{T}b, \frac{tT - t^2}{T}\right). \tag{8}$$

BBDM [28] develops an image-to-image translation method based on the Brownian Bridge process by treating $a$ and $b$ as two images. The forward diffusion process is defined as:

$$q(\mathbf{x}_t|\mathbf{x}_0, \mathbf{y}) = \mathcal{N}\left(\mathbf{x}_t; (1 - m_t)\mathbf{x}_0 + m_t\mathbf{y}, \delta_t\right), \tag{9}$$

$$\text{where } m_t = \frac{t}{T} \text{ and } \delta_t = 2s(m_t - m_t^2). \tag{10}$$

$\mathbf{x}_0$ and $\mathbf{y}$ are two images, and $s$ is a constant that controls the maximum variance in the Brownian Bridge. The sampling process is derived based on Bayes' theorem [28]:

$$p_\theta(\mathbf{x}_{t-1}|\mathbf{x}_t, \mathbf{y}) = q(\mathbf{x}_{t-1}|\mathbf{x}_0, \mathbf{x}_t, \mathbf{y})$$

$$= \frac{q(\mathbf{x}_t|\mathbf{x}_{t-1}, \mathbf{y})q(\mathbf{x}_{t-1}|\mathbf{x}_0, \mathbf{y})}{q(\mathbf{x}_t|\mathbf{x}_0, \mathbf{y})} \tag{11}$$

$$= \mathcal{N}(\tilde{\mu}_t, \tilde{\delta}_t \mathbf{I}).$$

where $\tilde{\mu}_t = c_{xt}\mathbf{x}_t + c_{yt}y + c_{\epsilon t}(m_t(\mathbf{y} - \mathbf{x}_0) + \sqrt{\delta_t}\epsilon)$,

$$c_{xt} = \frac{\delta_{t-1}}{\delta_t}\frac{1 - m_t}{1 - m_{t-1}} + \frac{\delta_{t|t-1}}{\delta_t}(1 - m_t),$$

$$c_{yt} = m_{t-1} - m_t\frac{1 - m_t}{1 - m_{t-1}}\frac{\delta_{t-1}}{\delta_t},$$

$$c_{\epsilon t} = (1 - m_{t-1})\frac{\delta_{t|t-1}}{\delta_t},$$

$$\delta_{t|t-1} = \delta_t - \delta_{t-1}\frac{(1 - m_t)^2}{(1 - m_{t-1})^2}.$$

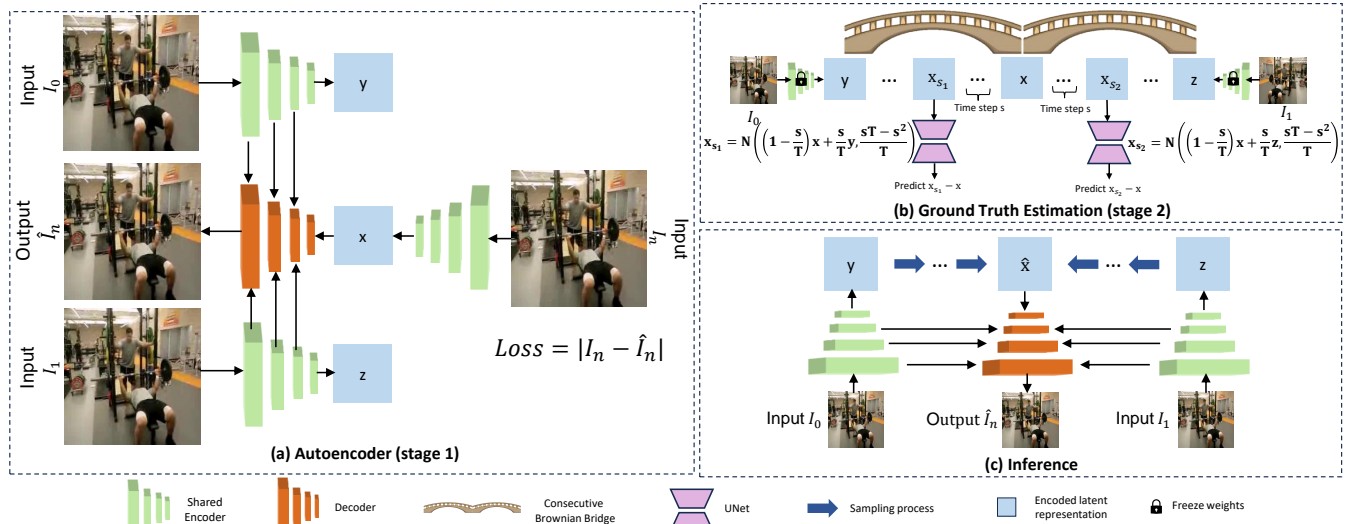

**Figure 2: The illustration of our two-stage method. The encoder is shared for all frames. (a) The autoencoder stage. In this stage, previous frame $I_0$, intermediate frame $I_n$, and next frame $I_1$ are encoded by the encoder to y, x, z respectively. Then x is fed to the decoder, together with the encoder feature of $I_0$, $I_1$ at different down-sampling factors. The decoder predicts the intermediate frame as $\hat{I}_n$. The encoder and decoder are trained in this stage. (b) The ground truth estimation stage. In this stage, y, x, z will be fed to the consecutive Brownian Bridge diffusion as three endpoints, where we sample two states that move time step $s$ from x in both directions. The UNet predicts the difference between the current state and x. The autoencoder is well-trained and frozen in this stage. (c) Inference. $\hat{x}$ is sampled from y, z to estimate x (details in Section 3.4). The decoder receives $\hat{x}$ and encoder features of $I_0$, $I_1$ at different down-sampling factors to interpolate the intermediate frame.**

It suffices to train a deep neural network $\epsilon_\theta$ to estimate the term $c_{\epsilon t}(m_t(y - x_0) + \sqrt{\delta_t}\epsilon)$, and therefore the training objective is $\mathbb{E}_{x_0,y,\epsilon}[c_{\epsilon t}||m_t(y - x_0) + \sqrt{\delta_t}\epsilon - \epsilon_\theta(x_t, t)||_2^2]$.

## 3.2 Formulation of Diffusion-based VFI

The goal of video frame interpolation is to estimate the intermediate frame $I_n$ given the previous frame $I_0$ and the next frame $I_1$. n is set to 0.5 to interpolate the frame in the middle of $I_0$ and $I_1$. In latent diffusion models [43], there is an autoencoder that encodes images to latent representations and decodes images from latent representations. The diffusion model is given a standard Gaussian noise, denoises it according to the sampling process, and decodes the denoised latent representation back to an image. Since the initial noise is random, the decoded images are diverse images when they are sampled repetitively with the same conditions such as poses. Instead of diversity, VFI looks for a deterministic ground truth, which is the intermediate frame. Such a ground truth frame is encoded to a ground truth latent representation by the encoder, and only the ground truth latent representation needs to be estimated since the decoder will decode it back to the frame. Therefore, LDM-based VFI can be split into two stages: autoencoder and ground truth estimation. The two stages are defined as:

(1) **Autoencoder**. The primary function of the autoencoder is similar to image compression: compressing images to latent representations so that the diffusion model can be efficiently implemented. We denote x, y, z as encoded latent representations of $I_n$, $I_0$, $I_1$. In this stage, the goal is to compress $I_n$ to x with an encoder and then reconstruct $I_n$ from x with a decoder. x is provided to the decoder together with neighboring frames $I_0$, $I_1$

and their features in the encoder at different down-sampling factors. The overview of this stage is shown in Figure 2 (a). However, to interpolate the intermediate frame, x is unknown, so we need to estimate this ground truth.

(2) **Ground truth estimation**. In this stage, the goal is to accurately estimate x with a diffusion model. The diffusion model converts x to y, z with the diffusion process, and we train a UNet to predict the difference between the current diffusion state and x, shown in Figure 2 (b). The sampling process of the diffusion model will convert y, z to x with the UNet output.

The autoencoder is modeled with VQModel [43] in Section 3.3, and the ground truth estimation is accomplished by our proposed (latent) consecutive Brownian Bridge diffusion in Section 3.4. During inference, both stages are combined as shown in Figure 2 (c), where we decode diffusion-generated latent representation $\hat{x}$. Via such formulation, we can have a more specific direction to improve VFI quality. If images decoded from x (Figure 2 (a)) have similar visual quality to images decoded from $\hat{x}$ (Figure 2 (c)), then the diffusion model achieves a strong performance in ground truth estimation, so it will be good to develop a good autoencoder. On the other way round, the performance of ground truth estimation can be potentially improved by redesigning the diffusion model.

## 3.3 Autoencoder

Diffusion models running in pixel space are extremely inefficient in video interpolation because videos can be up to 4K in real life [41]. Therefore, we can encode images into a latent space with encoder $\mathcal{E}$ and decode images from the latent space with decoder $\mathcal{D}$. Features of $I_0$, $I_1$ are included because detailed information may be lost when

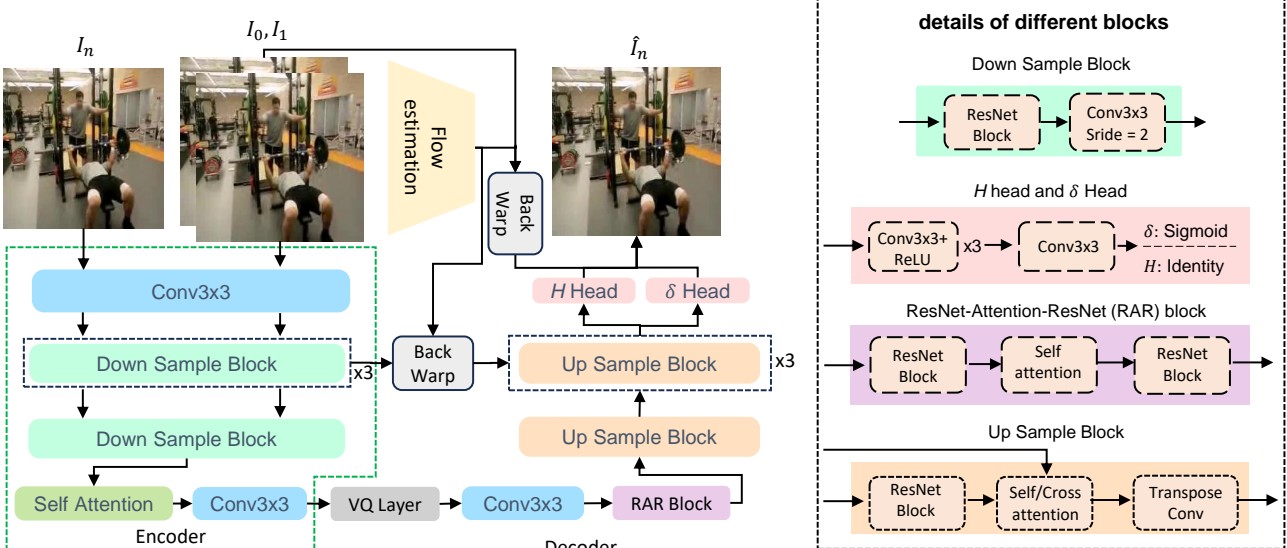

**Figure 3: Architecture of the autoencoder. The encoder is in green dashed boxes, and the decoder contains all remaining parts. The output of consecutive Brownian Bridge diffusion will be fed to the VQ layer. The features of $I_0, I_1$ at 2×, 4×, 8× down-sampling rate will be sent to the cross-attention module at Up Sample Block in the Decoder.**

images are encoded to latent representations [11]. We incorporate feature pyramids of neighboring frames into the decoder stage as guidance because neighboring frames contain a large number of shared details. Given $I_n, I_0, I_1$, the encoder $\mathcal{E}$ will output encoded latent representation $\mathbf{x}, \mathbf{y}, \mathbf{z}$ for diffusion models and feature pyramids of $I_0, I_1$ in different down-sampling rates, denoted $\{f_y^k\}, \{f_z^k\}$, where $k$ is down-sampling factor. When $k = 1$, $\{f_y^k\}$ and $\{f_z^k\}$ represent original images. The decoder $\mathcal{D}$ will take sampled latent representation $\hat{\mathbf{x}}$ (output of diffusion model that estimates $\mathbf{x}$) and feature pyramids $\{f_y^k\}, \{f_z^k\}$ to reconstruct $I_n$. In lines of equations, these can be expressed as:

$$\mathbf{x}, \mathbf{y}, \{f_y^k\}, \mathbf{z}, \{f_z^k\} = \mathcal{E}(I_n, I_0, I_1),$$
$$\hat{I}_n = \mathcal{D}\left(\mathbf{x}, \{f_y^k\}, \{f_z^k\}\right). \tag{12}$$

Our encoder shares an identical structure with that in LDMVFI [11], and we slightly modify the decoder to better fit the VFI task.

**Decoding with Warped Features.** LDMVFI [11] apply cross-attention [54] to up-sampled $\hat{\mathbf{x}}$ and $f_x^k, f_y^k$, but keeping feature of neighboring frames may preserve their original information (i.e. motion in previous and next frames). This is problematic since motion changes may be drastic in different frames. Therefore, we estimate optical flows from $I_n$ to $I_0, I_1$ with a flow estimation module and apply backward warping to the feature pyramids. Suppose $\hat{x}$ is generated by our consecutive Brownian Bridge diffusion, and it is up-sampled to $h^k$ where $k$ denotes the down-sampling factor compared to the original image. Then, we apply $CA\left(h^k, Cat(warp(f_y^k), warp(f_z^k))\right)$ for $k > 1$ to fuse the latent representation $h^k$ and feature pyramids $f_y^k$ and $f_z^k$, where $CA(\cdot, \cdot)$, $Cat(\cdot, \cdot)$, and $warp(\cdot)$ denotes cross attention, channel-wise concatenation, and backward warping with estimated optical flows respectively. Finally, we apply convolution layers to $h^1$ to

predict soft mask $H$ and residual $\delta$. The interpolation output is $\hat{I}_n = H * warp(I_0) + (1 - H) * warp(I_1) + \delta$, where $*$ holds for Hadamard product, and $\hat{I}_n$ is the reconstructed image. The detailed illustration of the architecture is shown in Figure 3. The VQ layer is connected with the encoder during training, but it is disconnected from the encoder and receives the sampled latent representation from the diffusion model.

## 3.4 Consecutive Brownian Bridge Diffusion

Brownian Bridge diffusion model (BBDM) [28] is designed for translation between image pairs, connecting two deterministic points, which seems to be a good solution to estimate the ground truth intermediate frame. However, it does not fit the VFI task. In VFI, images are provided as triplets because we aim to reconstruct intermediate frames giving neighboring frames, resulting in three points that need to be connected. If we construct a Brownian Bridge between the intermediate frame and the next frame, then the previous frame is ignored, and so is the other way round. This is problematic because we do not know what "intermediate" is if we lose one of its neighbors. Therefore, we need a process that transits among three images. Given two neighboring images $I_0, I_1$, we aim to construct a Brownian Bridge process with endpoints $I_0, I_1$ and additionally condition its middle stage on the intermediate frame $I_n$ ($n = 0.5$ for 2× interpolation). To achieve this, the process starts at $t = 0$ with value $\mathbf{y}$, passes $t = T$ with value $\mathbf{x}$, and ends at $t = 2T$ with value $\mathbf{z}$. To be consistent with the notation in diffusion models, $\mathbf{x}, \mathbf{y}, \mathbf{z}$ are used to represent latent representations of $I_n, I_0, I_1$ respectively. It is therefore defined as $X_t = W_t | W_0 = \mathbf{y}, W_T = \mathbf{x}, W_{2T} = \mathbf{z}$. The sampling process starts from time 0 and 2T and goes to time T. Such a process indeed consists of two Brownian Bridges, where the first one ends at $\mathbf{x}$ and the second one starts at $\mathbf{x}$. We can easily

**Algorithm 1** Training

1: **repeat**
2:     sample triplet $\mathbf{x}, \mathbf{y}, \mathbf{z}$ from dataset
3:     $s \leftarrow Uniform(0, T)$
4:     $w_s \leftarrow min\{\frac{1}{\delta_t}, \gamma\}$          ▷ $\gamma$ is a pre-defined constant
5:     $\epsilon \leftarrow \mathcal{N}(\mathbf{0}, \mathbf{I})$
6:     $\mathbf{x_{s_1}} \leftarrow \frac{s}{T}\mathbf{x} + (1 - \frac{s}{T})\mathbf{y} + \sqrt{\frac{s(T-s)}{T}}\epsilon$
7:     $\mathbf{x_{s_2}} \leftarrow \frac{s}{T}\mathbf{x} + (1 - \frac{s}{T})\mathbf{z} + \sqrt{\frac{s(T-s)}{T}}\epsilon$
8:     $\mathbf{r} \leftarrow Uniform(0, 1)$
9:     **if** r < 0.5 **then** take a gradient step on
10:        $\nabla_\theta ||\epsilon_\theta(\mathbf{x}_{s_1}, T - s, \mathbf{y}, \mathbf{z}) - (\mathbf{x}_{s_1} - \mathbf{x})||_2^2$
11:     **else** take a gradient step on
12:        $\nabla_\theta ||\epsilon_\theta(\mathbf{x}_{s_2}, T + s, \mathbf{y}, \mathbf{z}) - (\mathbf{x}_{s_2} - \mathbf{x})||_2^2$
13:     **end if**
14: **until** convergence

---

**Algorithm 2** Sampling

1: $t_1, t_2 \leftarrow T, \Delta_t \leftarrow \frac{T}{\text{sampling steps}}, \mathbf{x}_{T_1} = \mathbf{y}, \mathbf{x}_{T_2} = \mathbf{z}$
2: **repeat**
3:     $s_1, s_2 \leftarrow t_1 - \Delta_t, t_2 - \Delta_t$
4:     $\epsilon \leftarrow \mathcal{N}(\mathbf{0}, \mathbf{I})$
5:     $\mathbf{x_{s_1}} \leftarrow x_{t_1} - \frac{\Delta_t}{t_1}\epsilon_\theta(x_{t_1}, T - t_1, \mathbf{y}, \mathbf{z}) + \sqrt{\frac{s_1 \Delta_t}{t_1}}\epsilon$
6:     $\mathbf{x_{s_2}} \leftarrow x_{t_2} - \frac{\Delta_t}{t_2}\epsilon_\theta(x_{t_2}, T - t_2, \mathbf{y}, \mathbf{z}) + \sqrt{\frac{s_2 \Delta_t}{t_2}}\epsilon$
7:     $t_1, t_2 \leftarrow s_1, s_2$
8: **until** $t_1, t_2 = 0$

---

verify that for $0 < t < h$:

$$W_s | (W_0, W_t, W_h) = \begin{cases} W_s | (W_0, W_t) & \text{if } s < t \\ W_s | (W_t, W_h) & \text{if } s > t \end{cases}. \quad (13)$$

According to Eq. (13), we can derive the distribution of our consecutive Brownian Bridge diffusion (details shown in supplementary materials):

$$q(\mathbf{x}_t | \mathbf{y}, \mathbf{x}, \mathbf{z}) = \begin{cases} \mathcal{N}(\frac{s}{T}\mathbf{x} + (1 - \frac{s}{T})\mathbf{y}, \frac{s(T-s)}{T}\mathbf{I}) & s = T - t, t < T \\ \mathcal{N}(\frac{s}{T}\mathbf{x} + (1 - \frac{s}{T})\mathbf{z}, \frac{s(T-s)}{T}\mathbf{I}) & s = t - T, t > T \end{cases}. \quad (14)$$

**Cleaner Formulation.** Eq. (11) is in a discrete setup (i.e. time = $0, 1, ..., T$), and the sampling process is derived via Bayes' theorem, resulting in a complicated formulation. To preserve the maximum variance, it suffices to have $T = 2s$ in Eq. (8) and discretize T for training and sampling. Our forward diffusion is defined as Eq. (14). To sample from time $s$ from $t$ ($s < t$), we rewrite Eq. (11) according to Eq. (13):

$$p_\theta(\mathbf{x}_s | \mathbf{x}_t, \mathbf{y}) = q(\mathbf{x}_s | \mathbf{x}, \mathbf{x}_t, \mathbf{y})$$
$$= q(\mathbf{x}_s | \mathbf{x}, \mathbf{x}_t)$$
$$= \mathcal{N}\left(\mathbf{x}_s; \frac{s}{t}\mathbf{x}_t + (1 - \frac{s}{t})\mathbf{x}, \frac{s(t-s)}{t}\mathbf{I}\right) \quad (15)$$
$$= \mathcal{N}\left(\mathbf{x}_s; \mathbf{x}_t - \frac{t-s}{t}(\mathbf{x}_t - \mathbf{x}), \frac{s(t-s)}{t}\mathbf{I}\right).$$

Note that Eq. (11) is slightly different from ours in that it uses $\mathbf{x}_0$ to represent $\mathbf{x}$, but we directly use $\mathbf{x}$. Since we have a closed-form solution of $p_\theta(\mathbf{x}_s | \mathbf{x}_t, \mathbf{y})$ for $0 < s < t < T$, our method does not need DDIM [48] sampling for acceleration.

**Training and Sampling.** According to Eq. (15), it suffices to have a neural network $\epsilon_\theta$ estimating $\mathbf{x}_t - \mathbf{x}_0$. Moreover, based on Eq. (14), we can sample $s$ from $Uniform(0, T)$ and compute $t = T \pm s$ for $t > T$ and $T < t$. With one sample of $s$, we can obtain two samples at each side of our consecutive Brownian bridge diffusion symmetric at T. $\mathbf{y}, \mathbf{z}$ are added to the denoising UNet as extra conditions. Therefore, the training objective becomes:

$$\mathbb{E}_{\{\mathbf{y},\mathbf{x},\mathbf{z}\},\epsilon}[||\epsilon_\theta(\mathbf{x}_{s_1}, T - s, \mathbf{y}, \mathbf{z}) - (\mathbf{x}_{s_1} - \mathbf{x})||_2^2]$$
$$+ \mathbb{E}_{\{\mathbf{y},\mathbf{x},\mathbf{z}\},\epsilon}[||\epsilon_\theta(\mathbf{x}_{s_2}, T + s, \mathbf{y}, \mathbf{z}) - (\mathbf{x}_{s_2} - \mathbf{x})||_2^2]. \quad (16)$$

where $\mathbf{x_{s_1}} = \frac{s}{T}\mathbf{x} + (1 - \frac{s}{T})\mathbf{y} + \sqrt{\frac{s(T-s)}{T}}\epsilon$,

$$\mathbf{x_{s_2}} = \frac{s}{T}\mathbf{x} + (1 - \frac{s}{T})\mathbf{z} + \sqrt{\frac{s(T-s)}{T}}\epsilon, \quad (17)$$

$$\epsilon \sim \mathcal{N}(\mathbf{0}, \mathbf{I}).$$

Optimizing Eq. (16) requires two forward calls of the denoising UNet, so to be more efficient in computation, we randomly select one of them to optimize during training. Moreover, [15] proposes $min - SNR - \gamma$ weighting for different time steps during training based on the signal-to-noise ratio, defined as $min\{SNR(t), \gamma\}$. In DDPM [17], we have $SNR(t) = \frac{\alpha_t}{1 - \alpha_t}$ because the mean and standard deviation are scaled by $\sqrt{\alpha_t}$ and $\sqrt{1 - \alpha_t}$ respectively in the diffusion process. However, in our formulation, consecutive frames $I_0, I_1$ share almost identical mean, and so as their encoded latent representations. Therefore, the mean is never scaled down. The SNR is defined as $\frac{1}{\delta_t}$, where $\delta_t$ is the standard deviation of the diffusion process at time $t$. With the $min - SNR - \gamma$ weighting, the weighting of loss is defined as $w_t = min\{\frac{1}{\delta_t}, \gamma\}$.

The training algorithm is shown in Algorithm 1. To sample from neighboring frames, we can sample from either of the two endpoints $\mathbf{y}, \mathbf{z}$ with Eq. (14) and (15), shown in Algorithm 2. After sampling, we replace $\mathbf{x}$ in Eq (12) with the sampled latent representations to decode the interpolated frame.

**Cumulative Variance**. As we claimed, diffusion model [17] with conditional generation has a large cumulative variance while ours is much smaller. The cumulative variance for traditional conditional generation is larger than $1 + \sum_t \hat{\beta}_t$, which corresponds to 11.036 in experiments. However, in our method, such a cumulative variance is smaller than $T = 2$ in our experiments, resulting in a more deterministic estimation of the ground truth latent representations. The detailed justification is in the supplementary materials.

## 4 EXPERIMENTS

### 4.1 Implementations

**Autoencoder.** The down-sampling factor is set to be $f = 16$ for our autoencoder, which follows the setup of LDMVFI [11][1]. The flow estimation and refinement modules are initialized from pre-trained VFIformer [32] and frozen for better efficiency. The codebook size and embedding dimension of the VQ Layer are set to

---

[1]We follow their implementation and find that they achieve 16× down-sampling factor.

**Table 1: Quantitative results (LPIPS/FloLPIPS/FID, the lower the better) on test datasets. † means we evaluate our consecutive Brownian Bridge diffusion (trained on Vimeo 90K triplets [59] only) with autoencoder provided by LDMVFI [11]. The best performances are boldfaced, and the second best performances are underlined.**

| Methods | Middlebury | UCF-101 | DAVIS | SNU-FILM | | | |
|---|---|---|---|---|---|---|---|
| | | | | easy | medium | hard | extreme |
| | LPIPS/FloLPIPS/FID | LPIPS/FloLPIPS/FID | LPIPS/FloLPIPS/FID | LPIPS/FloLPIPS/FID | LPIPS/FloLPIPS/FID | LPIPS/FloLPIPS/FID | LPIPS/FloLPIPS/FID |
| ABME'21 [40] | 0.027/0.040/11.393 | 0.058/0.069/37.066 | 0.151/0.209/16.931 | 0.022/0.034/6.363 | 0.042/0.076/15.159 | 0.092/0.168/34.236 | 0.182/0.300/63.561 |
| MCVD'22 [55] | 0.123/0.138/41.053 | 0.155/0.169/102.054 | 0.247/0.293/28.002 | 0.199/0.230/32.246 | 0.213/0.243/37.474 | 0.250/0.292/51.529 | 0.320/0.385/83.156 |
| VFIformer'22 [32] | 0.015/0.024/9.439 | 0.033/0.040/22.513 | 0.127/0.184/14.407 | 0.018/0.029/5.918 | 0.033/0.053/11.271 | 0.061/0.100/22.775 | 0.119/0.185/40.586 |
| IFRNet'22 [26] | 0.015/0.030/10.029 | 0.029/0.034/20.589 | 0.106/0.156/12.422 | 0.021/0.031/6.863 | 0.034/0.050/12.197 | 0.059/0.093/23.254 | 0.116/0.182/42.824 |
| AMT'23 [29] | 0.015/0.023/**7.895** | 0.032/0.039/21.915 | 0.109/0.145/13.018 | 0.022/0.034/6.139 | 0.035/0.055/11.039 | 0.060/0.092/20.810 | 0.112/0.177/40.075 |
| UPR-Net'23 [24] | 0.015/0.024/7.935 | 0.032/0.039/21.970 | 0.134/0.172/15.002 | 0.018/0.029/5.669 | 0.034/0.052/10.983 | 0.062/0.097/22.127 | 0.112/0.176/40.098 |
| EMA-VFI'23 [60] | 0.015/0.025/8.358 | 0.032/0.038/21.395 | 0.132/0.166/15.186 | 0.019/0.038/5.882 | 0.033/0.053/11.051 | 0.060/0.091/**20.679** | 0.114/0.170/**39.051** |
| LDMVFI'24 [11] | 0.019/0.044/16.167 | **0.026**/0.035/26.301 | 0.107/0.153/12.554 | 0.014/0.024/5.752 | 0.028/0.053/12.485 | 0.060/0.114/26.520 | 0.123/0.204/47.042 |
| Ours† | 0.012/0.011/14.447 | 0.030/0.029/15.335 | 0.097/0.145/12.623 | 0.011/0.011/5.737 | 0.028/0.028/12.569 | 0.051/0.053/25.567 | 0.099/0.103/46.088 |
| Ours | **0.007**/**0.008**/7.964 | 0.029/**0.028**/14.022 | **0.052**/**0.086**/**10.170** | **0.010**/**0.010**/5.166 | **0.022**/**0.023**/9.571 | **0.035**/**0.035**/20.713 | **0.075**/**0.075**/41.545 |

16384 and 3 respectively. The number of channels in the compact latent space (encoder output) is set to 8. A self-attention [54] is applied at 16× down-sampling latent representation (both encoder and decoder), and cross attentions [54] with warped features are applied on 2×, 4×, and 8× down-sampling factors in the decoder. Following LDMVFI, max-attention [53] is applied in all attention layers for better efficiency. The model is trained with Adam optimizer [25] with a learning rate of $10^{-5}$ for 100 epochs with a batch size of 16. The autoencoder is still slowly converging after 100 epochs, but we stopped training to evaluate it.

**Consecutive Brownian Bridge Diffusion.** We set $T = 2$ (corresponding to maximum variance $\frac{1}{2}$) and discretize 1000 steps for training and 50 steps for sampling. The denoising UNet takes the concatenation of $x_t, y, z$ as input and is trained with Adam optimizer [25] with $10^{-4}$ learning rate for 30 epochs with a batch size of 64. $\gamma$ is set to be 5 in the $min - SNR - \gamma$ weighting.

## 4.2 Datasets and Evaluation Metrics

**Training Sets.** To ensure a fair comparison with most recent works [1, 7, 12, 18, 20, 24, 32, 34, 42, 47], we train our models in Vimeo 90K triplets dataset [59], which contains 51,312 triplets. We apply random flipping, random cropping to 256 × 256, temporal order reversing, and random rotation with multiples of 90 degrees as data augmentation.

**Test Sets.** We select UCF-101 [50], DAVIS [41], SNU-FILM [8], and Middlebury [2] to evaluate our method. UCF-101 and Middlebury consist of relatively low-resolution videos (less than 1K), whereas DAVIS and SNU-FILM consist of relatively high-resolution videos (up to 4K). SNU-FILM consists of four categories with increasing levels of difficulties (i.e. larger motion changes): easy, medium, hard, and extreme.

**Evaluation Metrics.** Recent works [10, 11, 62] reveal that PSNR and SSIM [56] are sometimes unreliable because they have relatively low correlation with humans' visual judgments. However, deep-learning-based metrics such as FID [16], LPIPS [62], and FloLPIPS [10] are shown to have a higher correlation with humans' visual judgments in [11, 62]. Moreover, in our experiments, we also find such inconsistencies between PSNR/SSIM and visual quality, which will be discussed in Section 4.3. Therefore, we select FID, LPIPS, and FloLPIPS as our main evaluation metrics. LPIPS

and FID measure distances in the space of deep learning features. FloLPIPS is based on LPIPS but takes the motion in the frames into consideration. Our methods evaluated with PSNR and SSIM will be included in the supplementary materials.

## 4.3 Experimental Results

**Quantitative Results.** Our method is compared with recent open-source state-of-the-art VFI methods, including ABME [40], MCVD [55], VFIformer [32], IFRNet [26], AMT [29], UPR-Net [24], EMA-VFI [60], and LDMVFI [11]. The evaluation is reported in LPIPS/FloLPIPS/FID (lower the better), shown in Table 1. We evaluate VFIformer, IFRNet, AMT, UPR-Net, and EMA-VFI with their trained weights, and other results are provided in the appendix of LDMVFI [11]. Models with different versions in the number of parameters are all chosen to be the largest ones. With the same autoencoder as LDMVFI [11], our method (denoted as ours†) generally achieves better performance than LDMVFI, indicating the effectiveness of our consecutive Brownian Bridge diffusion. Moreover, with an improved autoencoder, our method (denoted as ours) generally achieves state-of-the-art performance. It is important to note that we achieve much better FloLPIPS than other SOTAs, indicating our interpolated results achieve stronger motion consistency. In a few datasets, our method does not achieve the best performance in FID or LPIPS because our autoencoder is still converging.

**Qualitative Results.** In Table 1, our consecutive Brownian Bridge diffusion with the autoencoder in LDMVFI [11] (denoted as our†) generally achieves better quantitative results than LDMVFI, showing our method is effective. We include qualitative visualization in Figure 5 to support this result. Moreover, as mentioned in Section 1, we find that the autoencoder in [11] usually reconstructs overlaid images, and therefore we propose a new method of reconstruction. We provide examples to visualize the reconstruction results with our autoencoder and LDMVFI's autoencoder for comparison, shown in Figure 4. All examples are from SNU-FILM extreme [8], which contains relatively large motion changes in neighboring frames.

We have provided some visual comparisons of our method and recent SOTAs in Figure 1. Our method achieves better visual quality because we have clearer details such as dog skins, cloth with folds, and fences with nets. However, UPR-Net [24] achieves better PSNR/SSIM in all the cropped regions ($5 - 10\%$ better) than

**Table 2: Ablation studies of autoencoder and ground truth estimation. + GT means we input ground truth x to the decoder part of autoencoder. + BB indicates our consecutive Brownian Bridge diffusion trained with autoencoder of LDMVFI. With our consecutive Brownian Bridge diffusion, the interpolated frame has almost the same performance as the interpolated frame with ground truth latent representation, indicating the strong ground truth estimation capability. Our autoencoder also has better performance than LDMVFI [11].**

| Methods | Middlebury | UCF-101 | DAVIS | SNU-FILM | | | |
|---|---|---|---|---|---|---|---|
| | | | | easy | medium | hard | extreme |
| | LPIPS/FloLPIPS/FID | LPIPS/FloLPIPS/FID | LPIPS/FloLPIPS/FID | LPIPS/FloLPIPS/FID | LPIPS/FloLPIPS/FID | LPIPS/FloLPIPS/FID | LPIPS/FloLPIPS/FID |
| LDMVFI'24 [11] | 0.019/0.044/16.167 | 0.026/0.035/26.301 | 0.107 0.153/12.554 | 0.014/0.024/5.752 | 0.028/0.053/12.485 | 0.060/0.114/26.520 | 0.123 0.204/47.042 |
| LDMVFI'24 [11] + BB | 0.012/0.011/14.447 | 0.030/0.029/15.335 | 0.097/0.145/12.623 | 0.011/0.011/5.737 | 0.028/0.028/12.569 | 0.051/0.053/25.567 | 0.099/0.103/46.088 |
| LDMVFI'24 [11] + GT | 0.012/0.011/14.492 | 0.030/0.029/15.338 | 0.097/0.145/12.670 | 0.011/0.011/5.738 | 0.028/0.028/12.574 | 0.051/0.053/25.655 | 0.099/0.103/46.080 |
| Ours | 0.007/0.008/7.964 | 0.029/0.028/14.022 | 0.052/0.086/10.170 | 0.010/0.010/5.166 | 0.022/0.023/9.571 | 0.035/0.035/20.713 | 0.075/0.075/41.545 |
| Ours + GT | 0.007/0.008/7.965 | 0.029/0.028/14.022 | 0.052/0.086/10.170 | 0.010/0.010/5.166 | 0.022/0.023/9.570 | 0.035/0.035/20.712 | 0.075/0.075/41.544 |

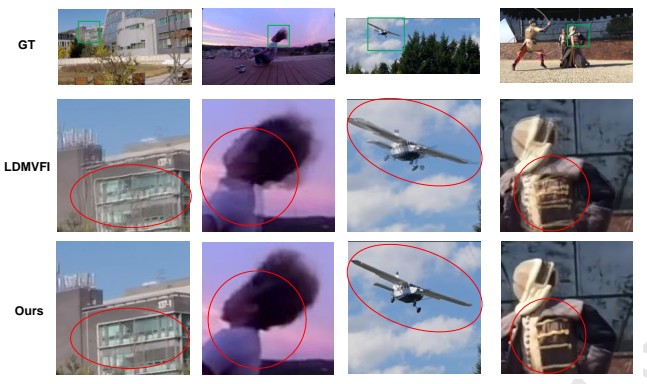

**Figure 4: The reconstruction quality of our autoencoder and LDMVFI's autoencoder (decoding with ground truth latent representation x). Images are cropped within green boxes for detailed comparisons. Red circles highlight the details that we have better reconstruction quality. LDMVFI usually outputs overlaid images while ours does not.**

ours, which is highly inconsistent with the visual quality. More qualitative results are provided in the supplementary materials.

## 4.4 Ablation Studies

As we discussed in Section 3.2, latent-diffusion-based VFI can be broken down into two stages, so we conduct an ablation study on the ground truth estimation capability of our consecutive Brownian Bridge diffusion. We compare the LPIPS/FloLPIPS/FID of decoded images with diffusion-generated latent representation $\hat{x}$ and ground truth $x$, which is encoded $I_n$. The results are shown in Table 2. It is important to note that, fixing inputs as the ground truth, our autoencoder achieves a stronger performance than the autoencoder in LDMVFI [11], indicating the effectiveness of our autoencoder. Also, fixing the autoencoder, our consecutive Brownian Bridge diffusion achieves almost identical performance with the ground truth, indicating its strong capability of ground truth estimation. However, the conditional generation model in LDMVFI [11] usually underperforms the autoencoder with ground truth inputs. Therefore, our method has a stronger ability in both the autoencoder and ground truth estimation stages. More ablation study is provided in the supplementary materials.

**Figure 5: The visual comparison of interpolated results of LDMVFI [11] vs our method with the same autoencoder in LDMVFI (LDMVFI vs our† in Table 1). With the same autoencoder, our method can still achieve better visual quality than LDMVFI, demonstrating the superiority of our proposed consecutive Brownian Bridge diffusion.**

## 5 CONCLUSION

In this study, we formulate the latent-diffusion-based VFI as a two-stage problem: autoencoder ground truth estimation. With this formulation, it is easy to figure out which part needs enhancements, guiding future research. We propose our consecutive Brownian Bridge diffusion that better estimates the ground truth latent representation due to its low cumulative variance. This method improves when the autoencoder is improved and achieves state-of-the-art performance with a simple yet effective design of the autoencoder, demonstrating its strong potential in VFI as a carefully designed autoencoder could potentially boost the performance by a large margin. Therefore, we believe our work will provide a unique research direction for diffusion-based frame interpolation.

**Limitations and Future Research.** Our method uses a bisection-like method to conduct multi-frame interpolation: we can interpolate $t = 0.5$ between $t = 0, 1$ and then interpolate $t = 0.25, 0.75$. However, our method cannot directly interpolate $t = 0.1$ from $t = 0, 1$. Future research can be conducted to resolve the limitations mentioned above or to improve autoencoders or diffusion models for better interpolation quality.

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
