# OpenReview forum: "Frame Interpolation with Consecutive Brownian Bridge Diffusion"
_acmmm.org/ACMMM/2024/Conference — MM2024 Poster_

### Official Review · Reviewer_xSZM · 2024-05-05

**Rating:** 5
**Confidence:** 3

**Summary:**

This paper introduces a novel method for video frame interpolation named Consecutive Brownian Bridge Diffusion.

The authors formulate the VFI problem into two stages: autoencoder and ground truth estimation and optimize each stage accordingly.

For the Autoencoder part, the authors use optical flow estimation and an additional loss for the enhancement, addressing issues related to reconstructing overlaid images.

For the Diffusion part, a Brownian Bridge Diffusion method is proposed. BBDM is previously used for distribution transformation, and a well-known downstream task: image-to-image translation.  The optimization process for Brownian Bridge alternates randomly. The sampling method is similar to Euler A.

Experimental results demonstrate that the method achieves state-of-the-art performance across multiple datasets and has the potential for further enhancement when the autoencoder is improved.

**Strengths:**

1. This work clearly and logically presents both its motivations and solutions. These include using optical flow estimation to optimize autoencoders and employing the Brownian bridge for frame interpolation, both of which are well-motivated.

2. The experimental section effectively demonstrates the efficacy of the proposed solutions. Table 1 shows that both the Brownian bridge interpolation method and the enhanced autoencoders have positive impacts.

3. The ablation study validates the effectiveness of individual modules.

**Limitations:**

There are some small concerns:

1. The author appears to need to compare several solutions involving the Schrödinger bridge. Many Schrödinger bridge-based approaches have been applied to image translation. Given this, is there also potential for them to be used in VFI? Could we briefly analyze the advantages and disadvantages between BBDM and Schrödinger Bridges?

2. The summary of related work on VAE seems to have some inaccuracies. LDM utilizing a VAE does not include a VQ module. However, the VQ module and codebook structure are present in dVAE of Taming Transformers (VQ-GAN).

There is one significant concern regarding the complexity of the proposed solution. What are the inference time and resource requirements? These details are not provided in the paper, yet they are crucial for VFI, which is particularly sensitive to inference latency. Authors could include a discussion of these aspects.

**Suitability:**

3

---

### Official Review · Reviewer_SGtP · 2024-05-24

**Rating:** 4
**Confidence:** 2

**Summary:**

In this submission, the authors approach the problem of latent-diffusion-based video frame interpolation (VFI) as a two-stage process involving autoencoder ground truth estimation. This framework clearly identifies areas needing enhancement, providing a roadmap for future research. They introduce the consecutive Brownian Bridge diffusion technique, which better estimates the ground truth latent representation due to its low cumulative variance. This method shows improvement when the autoencoder is refined and achieves state-of-the-art performance with a straightforward yet effective autoencoder design.

**Strengths:**

1. Clean and good formula, make it easy to read and follow.

2. The proposed method is based on some theories (Brownian Bridge), make this paper distinct from others.

3. Very clean illustration of the model architecture.

4. Extensive experiments results (both numerical and visualization) show its advantages.

**Limitations:**

1. The motivation is not well enough. It is claimed that "VFI expects that the output is deterministically", while the generated images of LDM is diverse. Well I think for every diffusion generation tasks, the training images is also deterministically to the GT. So the motivation seems unclear to me.

2. The background about stochastic process is missing in the related works.

3. Minor: The font size in figures should match the main paper.

**Suitability:**

2

---

### Official Review · Reviewer_FYFJ · 2024-05-24

**Rating:** 3
**Confidence:** 3

**Summary:**

The authors propose a new video frame interpolation (VFI) method based on Brownian Bridge Diffusion. Due to the reduced variance of Brownian bridge diffusion, and because of the triplet pair nature of VFI, they propose a novel consecutive Brownian bridge diffusion which operates on two endpoints with an additional middle point. The authors also propose changes, namely warping features from reference frames and injecting them via cross attention, to the latent diffusion autoencoder architecture for increased performance. These modifications result in better performance over a wide range of datasets.

**Strengths:**

* Interesting and novel idea with the consecutive brownian bridge diffusion
* Comprehensive experiments
* Good results

**Limitations:**

* I do not believe that formulating VFI as two stages, autoencoder and ground truth estimation, is a novel concept. LDMVFI [1] also has this two stage approach, and while this can provide motivation to improve method in two separate ways I don't think this is a novel contribution.
* I find the writing of the manuscript unclear, specifically section 3.2, and generally the flow of the manuscript was somewhat difficult to fully understand unless reading many times.
* The main motivation was to reduce the variance of diffusion sampling. Why is DDIM on standard diffusion models not sufficient? It is briefly touched on lines 145-147 but not backed up with any evidence, and the cumulative variance experiment in the supplemental section 2.2 is performed on DDPM, not DDIM.
* Additional video examples would be appreciated, as there are only a few shown in the supplementary material and this is a video frame interpolation method

[1] Danier, Duolikun, Fan Zhang, and David Bull. "Ldmvfi: Video frame interpolation with latent diffusion models." Proceedings of the AAAI Conference on Artificial Intelligence. Vol. 38. No. 2. 2024.

**Suitability:**

2

---

### Meta-Review · Area_Chair_wT2r · 2024-06-27

**Recommendation:** Accept (Poster)
**Confidence:** 4

**Metareview:**

The paper proposes a video frame interpolation method based on Brownian Bridge Diffusion. In particular, it proposes a consecutive Brownian bridge diffusion which operates on two endpoints with an additional middle point. It also warps features from reference frames and injects them via cross-attention, to the latent diffusion autoencoder architecture for increased performance. These modifications result in better performance over a wide range of datasets.

All reviewers agree that the proposed method is well-written and effective and the concerns are well-addressed during rebuttal.

I recommend acceptance.